# Association of Strength Performance in Bench Press and Squat with Anthropometric Variables between Resistance-Trained Males and Females

**DOI:** 10.3390/jfmk8010019

**Published:** 2023-02-01

**Authors:** Hallvard Nygaard Falch, Markus Estifanos Haugen, Stian Larsen, Roland van den Tillaar

**Affiliations:** Department of Sports Sciences and Physical Education, Nord University, 7600 Levanger, Norway

**Keywords:** strength training, sex, 1-RM performance

## Abstract

Individual differences in the appropriate percentage of 1-RM for a given repetition range could be a result of variation in anthropometrics and/or sex. Strength endurance is the term used to describe the ability to perform a number of repetitions prior to failure (AMRAP) in sub-maximal lifts and is important in determining the appropriate load for the targeted repetition range. Earlier research investigating the association of AMRAP performance and anthropometric variables was often performed in a sample of pooled sexes or one sex only or by utilizing tests with low ecological validity. As such, this randomized cross-over study investigates the association of anthropometrics with different measures of strength (maximal and relative strength and AMRAP) in the squat and bench press for resistance-trained males (*n* = 19, 24.3 ± 3.5 years, 182 ± 7.3 cm, 87.1 ± 13.3 kg) and females (*n* = 17, 22.1 ± 3 years, 166.1 ± 3.7 cm, 65.5 ± 5.6 kg) and whether the association differs between the sexes. Participants were tested for 1-RM strength and AMRAP performance, with 60% of 1-RM in the squat and bench press. Correlational analysis revealed that for all participants, lean mass and body height were associated with 1-RM strength in the squat and bench press (0.66, *p* ≤ 0.01), while body height was inversely associated with AMRAP performance (r ≤ −0.36, *p* ≤ 0.02). Females had lower maximal and relative strength with a greater AMRAP performance. In the AMRAP squat, thigh length was inversely associated with performance in males, while fat percentage was inversely associated with performance in females. It was concluded that associations between strength performance and anthropometric variables differed for males and females in fat percentage, lean mass, and thigh length.

## 1. Introduction

The benefits of resistance training are numerous, as incorporating habitual resistance training may increase muscular strength and muscular cross-sectional area, as well as improve markers of health, function in daily living [1,2,3], and sporting performance [4,5,6,7]. Enhancements in strength can be defined as an increased ability to exert force against external resistance [7], whereby strength again can be divided into several sub-categories, such as maximal strength (absolute force exertion), relative strength (force exerted per unit of body mass), and strength endurance (ability to resist fatigue and reductions in force output) [8]. Resistance training for improving sporting performance varies based on the distinctiveness and needs analysis for different sports. For example, in collision sports, there is a great focus on maximal strength (absolute force), despite increasing body mass, due to the importance of sprint momentum [9]. In team sports, however, where athletes have to accelerate their body mass, strength relative to their body mass seems like the more appropriate training goal [4,10,11]. In sports requiring multiple repetitions of a similar movement, such as Crossfit™, high levels of local muscular endurance are required of the athlete [12].

Males commonly possess higher levels of maximal and relative strength [13], mainly due to a greater amount of muscle mass and a lower body fat percentage [14,15], increasing the force capacity per unit of body mass. On the other hand, females are thought to outperform men in strength endurance tasks [8,15,16,17]. The sex-related differences observed in fatiguability when performing strength endurance work are not fully understood, but some mechanisms have been proposed. It could be a result of females possessing a greater proportion of fatigue-resistant type I fibers [15,18]. Sex differences are also studied in metabolism, suggesting males have a greater reliance on glycolytic pathways as opposed to females’ greater reliance on fat oxidation [8]. Another proposed mechanism is that lower absolute muscle force at a similar relative intensity leads to less intramuscular compression, thus enhancing oxygen availability and allowing easier clearance of metabolites through greater blood flow. In conjunction, less muscle mass would also reduce oxygen requirements [8]. From a mechanical perspective, shorter segment lengths lead to less work (force x distance) and external torque requirements (load x moment arm) per repetition being performed [13], a mechanical advantage for females due to a commonly shorter stature and length of limbs. The resistive torque and work requirements can be manipulated within exercises, such as by adjusting grip width in the bench press [19], although longer limbs are a disadvantage in theory.

The extent to which sex-related strength differences are observed is furthermore suggested to be task-specific [17,20], as the sex-related difference in maximal strength has been indicated to be greater in the upper limbs compared to the lower limbs [21,22]. Additionally, strength endurance performance favoring females seems to be greater in isometric tests, with work being performed at a lower percentage of maximal voluntary contraction (<80% of 1-RM) [8,16]. The aforementioned considerations are important when prescribing training programs, as resistance training programs often prescribe loads to athletes based on percentages of their 1-RM [23] to elicit specific adaptations. However, the maximal number of repetitions performed at a prescribed percentage of 1-RM may vary greatly between individuals based on anthropometrics, type of exercise, and sex of the athlete [17].

Furthermore, research investigating sex-related differences is often performed on untrained or moderately trained individuals [24]. It is therefore of interest to investigate if the association of anthropometric variables for maximal strength, relative strength, and strength endurance varies between males and females in a strength-trained population. Another gap in the literature is that sex-specific differences, especially for muscular endurance work, are commonly assessed in tests that lack ecological validity for what is being performed in training. As such, this study investigates the association of different anthropometric variables with measures of strength (maximal and relative strength and strength endurance) in the squat and bench press in strength-trained individuals matched in age and training experience and whether the association differs between the sexes. The study is valuable in comparing the association of anthropometric variables and strength performance between the sexes, as a conclusion based on a large mixed sample can lead to false conclusions since the results might be skewed due to anthropometric differences between the sexes (body height, body mass, and fat percentage). Females were hypothesized to have lower maximal and relative strength but greater strength endurance based on earlier research [8,17]. Within the sexes, lean mass and fat percentage were hypothesized to be associated with maximal and relative strength, with limb lengths inversely associated with strength endurance [13].

## 2. Materials and Methods

A randomized cross-over trial was performed to investigate the association between different anthropometric variables and strength in the squat and bench press for strength-trained male and female participants. To familiarize the participants with the testing procedure and establish levels of strength, all participants took part in a familiarization session > 72 h before the day of testing, consisting of an identical testing protocol. Participants were instructed not to train <24 h before testing and not to consume caffeine on the day of testing. They had to record a 24-h food log before the familiarization and were asked to replicate it to minimize variation in energy intake and hydration. The day of testing started with the participants’ height and segment lengths (upper arm, lower arm, thigh, and shank) being manually measured three times to the nearest 0.1 cm by a researcher, and at least two of the measurements had to be identical for the measurement to be valid. No measure violated this requirement, with the measurements being based upon hallmarks (upper arm: acromion to the lateral epicondyle of the distal part of the humerus; lower arm: lateral epicondyle of the distal part of the humerus to the lateral epicondyle of the distal part of the ulna; upper leg: greater trochanter of the femur to the distal lateral condyle of the femur; shank: distal lateral condyle of the femur to the lateral malleolus). Afterward, participants were weighed, with body composition estimated by a calibrated Tanita bioelectrical impedance device (MC-780MA). Then, the warm-up was initiated, which was performed in a similar manner for both the back squat and bench press. Stance and grip width were measured for the squat and bench press on the day of familiarization, which were required to be similar in the 1-RM and AMRAP tests. To avoid reductions in ecological validity, the participants were not constrained in the use of equipment such as chalk, belts, lifting shoes, or wrist wraps, as long as the equipment was kept similar through all trials.

### 2.1. Participants

A total of 36 resistance-trained males (*n* = 19) and females (*n* = 17) with no injury or illness negatively affecting performance in the squat and bench press participated in the study. The participants were required to be >18 years old with a minimum of 12 months of consistent resistance training with >2 sessions per week. Furthermore, participants had to be able to lift 1 and 1.2 × body mass in the bench press and squat for males and 0.7 and 1 × body mass in the bench press and squat for females. The study procedure was explained both orally and in writing, and written consent had to be signed before participation. This study was approved by the local ethics committee and the Norwegian Center for Research Data, and it conformed to the latest revision of the Helsinki Declaration (project No. 445723).

### 2.2. Testing

The warm-up started with the participants performing a self-selected number of repetitions with a 20-kg barbell (ata Powerbar stainless steel 29 mm, ata Group AS, Asker, Norway), followed by a standardized number of repetitions at different percentages of the estimated 1-RM (8 repetitions at 40%, 6 repetitions at 60%, 3 repetitions at 70%, and 2 repetitions at 80%) [25]. The participants subsequently performed 1-RM attempts, with load increments of 0.25 to 5 kg for every successful attempt after 4 min of rest, until true 1-RM was established. Load increments were conducted with calibrated (±10 g) plates, ranging from 0.25 kg to 50 kg (ata Powerlifting Steel Plate, ata Group AS, Asker, Norway). To complete a successful lift in the squat, the participant had to descend until the trochanter major was below the patella before initiating the ascending phase. In the bench press, the barbell had to descend until it touched the chest without bouncing before ascending until the elbows were fully extended. The feet, glutes, and upper back had to be in contact with the surface and the bench throughout the lift. The technical requirements were visually controlled by an experienced strength-and-conditioning professional, while two spotters secured safety in each lift.

The 1-RM on the day of testing in the squat and bench press was used to establish the load for the AMRAP test (60% of the 1-RM), which was performed after the 1-RM test with similar technical requirements. No rest was allowed in the AMRAP test, whereby too long a pause (>1 s in the top position of the lift) resulted firstly in a warning, while a second pause of >1 s was defined as failure. The participants performed as many repetitions as possible until they were unable to complete a full repetition without assistance from the spotters.

### 2.3. Statistical Analysis

Descriptive statistics are presented as means and standard deviations. The intraclass correlation coefficient (ICC) from the familiarization day to the test day was calculated to investigate the reliability in the squat and bench press when performing the 1-RM and AMRAP tests, in which the interpretation of the ICC was that values between 0.5 and 0.75 indicated moderate reliability, between 0.75 and 0.9 good reliability, and above 0.9 excellent reliability [26]. Between-group differences were tested by the independent samples t-test. The assumption of normality was assessed with the Shapiro–Wilk test. When the assumption of normality was violated, the non-parametric Mann–Whitney U test was used. Between-group effects were calculated according to Cohen’s d ($\frac{M1-M2}{Pooled\;STD}$). Effect sizes were defined as follows: 0.01 to 0.2 = very small; 0.2 to 0.5 = small; 0.5 to 0.8 = moderate; >0.8 = large; >1.2 = very large; and >2 = huge [27,28]. The correlation between performance and anthropometric variables was calculated with Pearson’s correlation coefficient. When the assumption of normality was violated, Spearman’s rho was used. The strength of association was defined by the following *r* value: 0.1 to 0.3 = small; 0.3 to 0.5 = moderate; 0.5 to 0.7 = large; and 0.7 ≥ very large. The Holm–Bonferroni correction was assessed to reduce the type I error rate for the number of correlational tests performed. The between-group difference in correlation coefficients was calculated by Fisher´s Z-test with an online calculator [29]. Relative strength in the squat and bench press was calculated as external load lifted/body mass (kg). All tests were performed in SPSS v.27 (IBM Corp., Armonk, NY, USA). The level of significance was set at *p* < 0.05.

## 3. Results

The ICC from the familiarization day to the test day revealed good-to-excellent reliability in the squat and bench press when performing the 1-RM and AMRAP tests (ICC ≥ 0.76). Significant differences were observed between males and females for all measures of anthropometrics and strength performance (d ≥ 0.87, *p* ≤ 0.05), except for age, training experience, and the number of repetitions in the AMRAP squat (d ≤ 0.68, *p* ≥ 0.07) (Table 1). Males were taller and heaver and had longer upper and lower limbs, a lower fat percentage, more lean mass, and higher absolute 1-RM performance in the squat and bench press than females. However, females had significantly more repetitions in the AMRAP bench press test than males (Table 1).

For all participants, lean mass revealed the greatest association with 1-RM performance in both the squat and bench press (r ≥ 0.81, *p* ≤ 0.01). Furthermore, body height was associated with increased 1-RM in the squat and bench press and relative strength in the bench press (r ≥ 0.55, *p* ≤ 0.01), but it was inversely associated with AMRAP performance in both the squat and bench press (r ≤ −0.36, *p* ≤ 0.02) (Table 2).

When analyzing between the sexes, significantly different correlation coefficients were observed between males and females for fat percentage in 1-RM strength and relative strength in the bench press and squat, lean mass in relative strength for the bench press, and thigh length in the AMRAP test for the squat (Z-score ≥ −0.198, *p* ≤ 0.05) (Table 2). Males increased absolute and relative 1-RM squat and bench press performance with increasing fat percentage, while women increased this performance with decreasing fat percentage (Figure 1).

Furthermore, females showed a positive correlation between increased lean body mass and AMRAP squat and relative 1-RM bench press performances, while males showed a negative correlation with these two parameters. In addition, the number of repetitions in the AMRAP squat test increased when thigh length was shorter in males, while no correlation was found in females (Figure 2).

## 4. Discussion

The current study aimed to compare the associations of anthropometric variables with measures of strength between resistance-trained males and females, matched in chronological age and training experience. It was hypothesized that females possessed lower maximal and relative strength and greater strength endurance, and limb lengths were hypothesized to be inversely associated with strength endurance within the sexes. The hypotheses were only partially confirmed. Unsurprisingly, males were significantly stronger in the squat and bench press in both maximal and relative strength. The differences in stature and body composition between the sexes most likely explain the strong association in all participants for height and lean mass with 1-RM strength. Earlier research suggests that heavier individuals with more muscle mass are generally stronger [30], and the males in this study were taller and heavier with more lean body mass in comparison to the females. Body height differences between males and females may also account for the inverse relationship between height and the AMRAP tests for all participants, as females performed significantly more repetitions in the AMRAP bench press and showed a trend towards significantly more repetitions in the AMRAP squat.

A significantly different correlation coefficient was observed for males and females between fat percentage and 1-RM performance and relative strength in both the squat and bench press. The results suggest increased fat percentage is associated with greater 1-RM strength in males but reduced relative strength in females. Increased fat percentage may contribute to the 1-RM bench press as a larger body mass could make the range of motion for the lift shorter (less work) by reducing bar path displacement from lockout to the bottom position of the lift. In the squat, the extra mass may aid in stabilizing the bar [30]. Unexpectedly, fat percentage in females trended towards being inversely associated with 1-RM strength, contradicting the association observed in males and earlier research [7,30]. Fat percentage in females was furthermore observed to be inversely associated with relative strength for both the squat and the bench press, as increasing body mass is suggested to increase the absolute weight lifted while decreasing relative strength [31], especially if mass increases are composed of non-contractile tissue.

Although greater lean mass is known to be associated with increased strength in powerlifting [32], a significantly different correlation coefficient was observed for males and females between lean mass and relative strength in the bench press, indicating lean mass to be a greater predictor of relative bench press strength in females compared to males. The finding was unexpected, as force per unit of muscle mass has been suggested to be similar between males and females [33]. Why this difference is observed cannot be stated for certain, although force per unit of muscle mass could be lower with increased muscle mass [33] and the difference in muscle mass between the sexes is commonly greater in the upper body [21]. Furthermore, lean mass was only significantly associated with increased 1-RM strength for all participants and females in the current study (Table 2). The trivial correlation in males may be due to sample size and variance in anthropometric variables, although variations in experience with the squat and bench press cannot be discounted as a confounding variable as neurological adaptations may affect the association between lean mass and 1-RM strength [33].

The different associations in strength performance with anthropometric variables observed between sexes are possibly a result of different training motives [34,35]. It could be that the female participants with lower fat percentages are also most devoted to resistance training, as weight management is one of several motivational factors for habitual resistance training in females [36,37]. In males, on the other hand, “being strong” is a traditionally valued trait [34], which at a certain point may come at the cost of increased body mass to further increase maximal strength [31]. Furthermore, females have been suggested to be more aware of self-presentation [35], whereby a certain “ideal” body composition might be a result of sex expectations [34]. The assumption of different motives is further supported by fat percentage being inversely associated with AMRAP squat performance in females (Table 2), as earlier research has associated repeated squat performance with VO2max while being inversely associated with body mass and fat percentage [38]. Therefore, the different associations between the sexes might be a result of training history [33]. However, these interpretations must be evaluated with caution, as training status, loading ranges, and type of exercise are all factors that may influence the observed association [20].

A significantly different correlation coefficient was observed for males and females between thigh length and the number of repetitions in the AMRAP squat (Figure 2). The results suggest that increased thigh length is a greater predictor of a decreased number of repetitions performed in the AMRAP squat for males compared to females, although an inverse association was expected in both sexes based on earlier research [13], as long femurs will increase the work performed per repetition. The difference might be a result of the males lifting greater absolute loads in the squat along with their greater body mass (Table 1). Thus, the absolute external torque requirements will be greater in males, possibly making each sub-maximal repetition relatively more fatiguing. This was not observed in the bench press, as the lengths of the upper and lower arm were not significantly associated with bench press performance, a relationship that has been found to vary in the literature based on the population studied [30]. Lastly, grip width relative to height was significantly associated with the number of repetitions performed by females in the AMRAP bench press. Increasing grip width might therefore be beneficial for females in the AMRAP bench press, possibly as a result of reducing work per repetition [19].

This study has some limitations that must be addressed. Firstly, the study would benefit from a larger sample size, as several moderate-to-large, non-significant r values were observed. A replication with a larger sample size is warranted. Secondly, lean mass accounts for all fat-free mass, such as water content and bone density. Therefore, individuals with greater body height/segment lengths might also have greater lean mass without having more contractile tissue. Thirdly, this study would be strengthened with the measurement of explanatory variables such as work being performed, cross-sectional area, and VO2max. Such measurements are warranted in future research to provide greater certainty in cause and effect. Lastly, this study used loads of 60% of 1-RM and might not be generalizable to other loading ranges.

## 5. Conclusions

Although cause and effect cannot be stated, fat percentage in females and thigh length in males are anthropometric variables indicated to influence appropriate loading ranges in the squat. It could be beneficial to evaluate these considerations in resistance training protocols when utilizing the barbell back squat for a strength-specific training goal. If maximal strength in the bench press is the training goal, males might benefit from increasing body mass. Increased bench press performance for females in this study was associated with greater lean mass but not increased fat percentage. However, it is important to underline that the sex-specific differences observed in this study may be influenced by the small sample size and training histories between the sexes. As such, the findings from this study should encourage caution when using small samples of mixed sexes to determine associations between strength and anthropometric variables.

## Figures and Tables

**Figure 1 jfmk-08-00019-f001:**
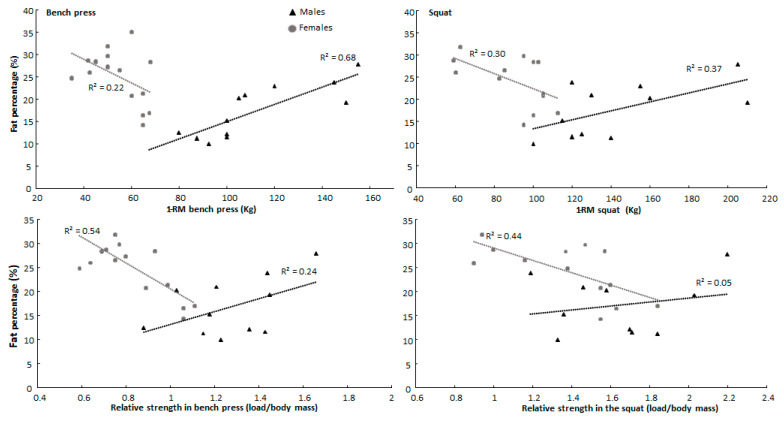
Sex-specific association between fat percentage with measures of maximal and relative strength in squat and bench press.

**Figure 2 jfmk-08-00019-f002:**
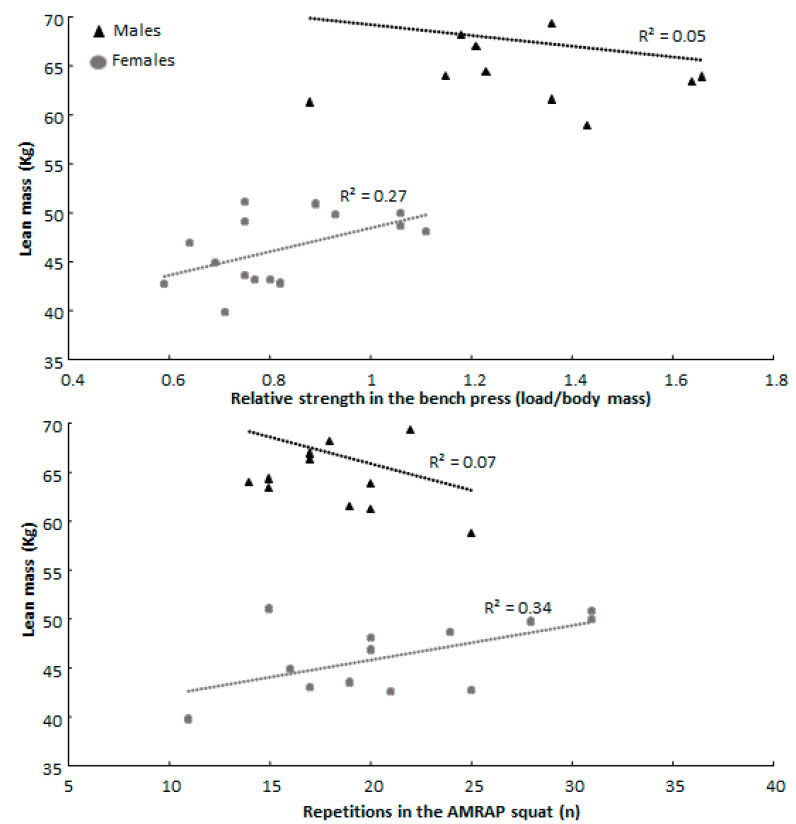
Association of lean mass and thigh length with relative strength in the bench press and the number of repetitions performed in the AMRAP squat for male and female participants.

**Table 1 jfmk-08-00019-t001:** Descriptive statistics of the male and female participants.

	Males (*n* = 19)	Females (*n* = 17)	Difference (%)	Effect Size (*d*)
Age (years)	24.3 ±3.5	22.1 ± 3	9.3	0.70
Height (cm)	182 ± 7.3	166.1 ± 3.7	8.7 *	2.87
Body mass (kg)	87.1 ± 13.3	65.5 ± 5.6	24.8 *	2.29
Training years (n of years)	4.2 ± 2.5	4.7 ± 2.3	10.8	0.21
Lean mass (kg)	67.4 ± 6.1	46.3 ± 3.6	31.3 *	4.35
Fat percentage (%)	17.3 ± 6	25.1 ± 6	31.1 *	1.30
Upper arm length (cm)	33.5 ± 3.2	30.7 ± 30.4	8.6 *	0.87
Lower arm length (cm)	28.9 ± 6.7	25.1 ± 1.6	13.4 *	0.93
Thigh length (cm)	42.4 ± 2.4	38.9 ± 3.2	8.3 *	1.28
Shank length (cm)	43.9 ± 2.6	39.7 ± 1.7	9.6 *	1.97
1-RM bench press (kg)	110.7 ± 24.3	54.6 ± 10	50.7 *	3.27
AMRAP bench press (n)	17.7 ± 2.6	20.9 ± 3.5	15.6 *	1.06
Relative strength bench press	1.3 ± 0.2	0.8 ± 0.2	34.8 *	2.25
1-RM squat (kg)	146 ± 34.9	88.6 ± 17.3	39.3 *	2.20
AMRAP squat (n)	18.4 ± 3.2	21.4 ± 5.8	14.2	0.68
Relative strength squat	1.7 ± 0.3	1.4 ± 0.3	19.2 *	1.14

* indicates a significant difference between males and females at a *p* < 0.05 level.

**Table 2 jfmk-08-00019-t002:** Correlations between different performances in the squat and bench press with anthropometric data for all male and female participants.

All Participants							
	Bench Press				Barbell Back Squat		
	1-RM	AMRAP	Relative Strength		1-RM	AMRAP	Relative Strength
Lean mass	0.86 *	−0.25	0.72 *	Lean mass	0.81 *	−0.23	0.42
Fat percentage	−0.49 *	0.34	0.52 *	Fat percentage	−0.25	−0.10	−0.41 *
Body height	0.75 *	−0.36	0.55 *	Body height	0.66 *	−0.41 *	0.24
Grip width/height	0.06	0.39	0.52	Stance width/height	0.15	−0.41 *	−0.03
Upper arm length	0.21	−0.02	0.06	Thigh length	0.46 *	−0.18	0.19
Lower arm length	0.40 *	0.01	0.11	Shank length	0.64 *	−0.26	0.43 *
**Males**							
	**1-RM**	**AMRAP**	**Relative Strength**		**1-RM**	**AMRAP**	**Relative Strength**
Lean mass	0.29	0.03	−0.21 †	Lean mass	0.46	−0.27 †	−0.01
Fat percentage	0.82 *†	0.07	0.49 †	Fat percentage	0.61 *†	0.08	0.22 †
Body height	−0.16	0.06	−0.59 *	Body height	0.09	−0.50	−0.42
Grip width/height	0.04	0.14	0.07	Stance width/height	−0.01	0.21	0.09
Upper arm length	−0.42	−0.18	−0.41	Thigh length	0.33	−0.67 *†	0.07
Lower arm length	−0.04	0.28	−0.22	Shank length	0.23	−0.48	−0.16
**Females**							
	**1-RM**	**AMRAP**	**Relative Strength**		**1-RM**	**AMRAP**	**Relative Strength**
Lean mass	0.75 *	0.49	0.52 †	Lean mass	0.57 *	0.58 †	0.39
Fat percentage	−0.47 †	0.13	−0.74 *†	Fat percentage	−0.55 †	−0.60 *	−0.67 *†
Body height	0.28	0.27	−0.08	Body height	−0.05	−0.07	−0.17
Grip width/height	0.48	0.59 *	0.43	Stance width/height	−0.47	0.02	−0.53
Upper arm length	−0.09	0.36	−0.24	Thigh length	0.03	0.06 †	0.19
Lower arm length	0.14	0.37	−0.23	Shank length	0.33	0.14	0.32

* indicates a significant correlation coefficient at a *p* < 0.05 level. † indicates a significantly different correlation coefficient between males and females at a *p* < 0.05 level.

## Data Availability

The raw data supporting the conclusion of this article will be made available by the authors without undue reservation.

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
