# Peer review of "Association of Strength Performance in Bench Press and Squat with Anthropometric Variables between Resistance-Trained Males and Females"

_jfmk, 2023, doi:10.3390/jfmk8010019_

Round 1

Reviewer 1 Report

This is an excellent article that is well written and of high relevance to the audience and field of sports science. I have the following minor comments and suggestions to the authors:

- Method: Last sentence of 2.2 Testing. It appears that you didn't give instructions on lifting velocity (e.g. lifting with maximal velocity as in velocity-based strength training). As such, the reference to 'standardized pre-determined tempo' seems misleading and I recommend to reconsider this statement.

- Results: If possible, I suggest to also analyse and present the association between body mass and strength-related parameters, as well as relative thigh length (e.g. divided by body mass or body height) and strength-related parameters. In particular, this may reveal further insights to help explain some of the observed differences between females and males.

- Discussion: Second paragraph. I recommend to reconsider the assumption that increased fat percentage in males is associated with increased chest circumference, and thus, reduced range of motion for bench press lifting.  In particular, you didn't measure chest circumference. Isn't it possible to have increased fat percentage but the same or even smaller chest circumference? 

- Conclusion: I recommend to significantly revise the conclusion. Currently, you are repeating the results. From my perspective, I wonder and would like to know How does your study advance the state-of-the-art? Possibly also put in relation to background information in introduction (e.g. benefits, needs analysis for different sports). What is the implication of your findings for strength training practice?

Author Response

Thank you for reviewing the manuscript. We have colored the changes in the manuscript.

This is an excellent article that is well written and of high relevance to the audience and field of sports science. I have the following minor comments and suggestions to the authors:

- Method: Last sentence of 2.2 Testing. It appears that you didn't give instructions on lifting velocity (e.g. lifting with maximal velocity as in velocity-based strength training). As such, the reference to 'standardized pre-determined tempo' seems misleading and I recommend to reconsider this statement.

The following part of the sentence has now been deleted: “…or failing to complete the subsequent repetition within the standardized pre-determined tempo”

- Results: If possible, I suggest to also analyse and present the association between body mass and strength-related parameters, as well as relative thigh length (e.g. divided by body mass or body height) and strength-related parameters. In particular, this may reveal further insights to help explain some of the observed differences between females and males.

We have checked all this, but did not find any associations. To keep the text to a short and to keep the story focused we did not include this to the text to avoid too much extra information that will make the story more difficult to follow.

- Discussion: Second paragraph. I recommend to reconsider the assumption that increased fat percentage in males is associated with increased chest circumference, and thus, reduced range of motion for bench press lifting.  In particular, you didn't measure chest circumference. Isn't it possible to have increased fat percentage but the same or even smaller chest circumference? 

The sentence has been rephrased: “Increased fat percentage may contribute to bench press 1-RM as a larger body mass could make the range of motion for the lift shorter (less work) by reducing bar path displacement from lock-out to the bottom position of the lift.”

- Conclusion: I recommend to significantly revise the conclusion. Currently, you are repeating the results. From my perspective, I wonder and would like to know How does your study advance the state-of-the-art? Possibly also put in relation to background information in introduction (e.g. benefits, needs analysis for different sports). What is the implication of your findings for strength training practice?

Conclusion has been revised: “Although cause- and effect cannot be stated, fat percentage in females and thigh lengths in males are anthropometric variables indicated to influence appropriate loading ranges in the squat. It could be beneficial to evaluate these considerations in resistance training protocols when utilizing the barbell back squat for a strength specific training goal. If maximal strength in bench press is the training goal, males might benefit from increasing body mass. Increased bench press performance for females in this study was associated with greater lean mass, but not increased fat percentage. However, it is important to underline that the sex specific differences observed in this study may be influenced by a small sample size and training history between the sexes. As such, the findings from this study should encourage caution when using small samples of mixed sexes to determine association between strength- and anthropometric variables.”

Reviewer 2 Report

General Comments:

The authors conducted a study to examine the influence of anthropometric and sex-related factors to AMRAP performance @60% of 1RM of the barbell bench press and barbell back squat exercises. The findings of the study are interesting not only for individuals working in traditional sports but CrossFit and tactical/occupational human performance. There are a few details which can be added to clarify aspects of the study to the reader. Also, with the AMRAP testing occurring on the same day after the 1RM testing it should be state the amount of rest between the tests and to acknowledge the potential effects of post-activation potentiation on the results in the present study.

Specific Comments

Abstract

-In the second sentence it would help for the sake of consistency with the introduction to specifically mention ‘strength endurance’ instead of immediately introducing AMRAP.

Introduction

-Generally the introduction was well-written, introduced relevant background knowledge and led to the aims of the study. However, if the authors could provide additional rationale for 1) investigating strength endurance at 60% of 1RM and 2) comparison of an upper body vs. lower body exercise it would be beneficial.

Methods

-Did familiarization and testing sessions take place at the same time of day to control fo diurnal variations in performance? If no this should be acknowledged as a potential limitation.

-Were knee sleeves/wraps permitted? Given the potential performance benefits of knee sleeves and belts for the squat exercise some additional detail (if recorded) on how frequently these were used is needed.

-Very good prescribing the tempo for the AMRAP testing. Too often this is not well-controlled in studies.  

-A flow diagram/schematic of experimental procedures would be beneficial to add. Particularly, to illustrate that the AMRAP testing took place on the SAME day as the 1RM.

-Line 158 – instead of saying assumption of normality was ‘controlled for’ by the Shapiro-Wilk test it would be better to say ‘assessed with’ the Shapiro-Wilk test.

Results

-The results are well-presented in my perspective.  

Discussion

-The authors make valid points in the discussion and offer several potential explanations for the findings. I believe the discussion can be improved by addressing some of the additional limitations in my earlier comments. Also, future studies could compute the work performed (lines 57-60 of introduction) during AMRAP testing to compare for differences or control for as a confounding variable.

Tables

-Table 2 – please add in the footnote that * indicates a significant correlation

Author Response

General Comments:

The authors conducted a study to examine the influence of anthropometric and sex-related factors to AMRAP performance @60% of 1RM of the barbell bench press and barbell back squat exercises. The findings of the study are interesting not only for individuals working in traditional sports but CrossFit and tactical/occupational human performance. There are a few details which can be added to clarify aspects of the study to the reader. Also, with the AMRAP testing occurring on the same day after the 1RM testing it should be state the amount of rest between the tests and to acknowledge the potential effects of post-activation potentiation on the results in the present study.

Specific Comments

Abstract

-In the second sentence it would help for the sake of consistency with the introduction to specifically mention ‘strength endurance’ instead of immediately introducing AMRAP.

The second sentence has been revised as follows: Strength endurance is the term used to describe the ability to perform number of repetitions prior to failure (AMRAP) in sub-maximal lifts and is important in determining appropriate load for targeted repetition range.

Introduction

-Generally the introduction was well-written, introduced relevant background knowledge and led to the aims of the study. However, if the authors could provide additional rationale for 1) investigating strength endurance at 60% of 1RM and 2) comparison of an upper body vs. lower body exercise it would be beneficial.

The rationale is already partially addressed; however, it can be discussed further if the reviewer think it is necessary:

  • “Also, strength endurance performance favoring females seems to be greater in isometric tests with work being performed at a lower percentage of maximal voluntary contraction (<80% of 1-RM) [8, 16].” As such, we wanted the load to be low enough to detect a difference if it exists (<80% of 1-RM). However, if the load was to light, other factors such as physiological factors may confound the results.

  • The extent to which sex-related strength differences are observed is furthermore suggested to be task specific [17, 20], as the sex-related difference in maximal strength has been indicated to be greater in the upper limbs compared to the lower limbs [21, 22].

Methods

-Did familiarization and testing sessions take place at the same time of day to control fo diurnal variations in performance? If no this should be acknowledged as a potential limitation.

Familiarization and testing sessions were performed at the same time of the day.

-Were knee sleeves/wraps permitted? Given the potential performance benefits of knee sleeves and belts for the squat exercise some additional detail (if recorded) on how frequently these were used is needed.

This is addressed in 2. Materials and Methods: “To avoid reductions in ecological validity, the participants were not constrained in the use of equipment such as chalk, belt, lifting shoes, or wrist wraps, as long as the equipment was kept similar through all trials.”

-Very good prescribing the tempo for the AMRAP testing. Too often this is not well-controlled in studies.  

Thank you.

-A flow diagram/schematic of experimental procedures would be beneficial to add. Particularly, to illustrate that the AMRAP testing took place on the SAME day as the 1RM.

We hope this is already clearly explained in the method section: “The 1-RM on the day of testing in the squat and bench press was used to establish the load for the AMRAP test (60% of 1-RM), which was performed after the 1-RM test with similar technical requirements.”
As this study is a within-subject design, with a 1-RM (maximal strength) followed by the AMRAP test (strength-endurance), we believe the effect of fatigue/PAP would be neglectable. Thus, we think a schematic is not necessary. We hope the reviewer agrees with our point of view but will add a schematic if the reviewer still thinks it is necessary.

-Line 158 – instead of saying assumption of normality was ‘controlled for’ by the Shapiro-Wilk test it would be better to say ‘assessed with’ the Shapiro-Wilk test.

Changed.

Results

-The results are well-presented in my perspective.  

Thank you.

Discussion

-The authors make valid points in the discussion and offer several potential explanations for the findings. I believe the discussion can be improved by addressing some of the additional limitations in my earlier comments. Also, future studies could compute the work performed (lines 57-60 of introduction) during AMRAP testing to compare for differences or control for as a confounding variable.

No computation of work being performed is now added as a limitation of the study.

Tables

-Table 2 – please add in the footnote that * indicates a significant correlation

Added.

Round 2

Reviewer 2 Report

The authors addressed most of the revisions and have conducted a nice study. I would still encourage them to consider adding a figure to visually present the protocol to the readers. Their will be many practitioners, and potentially students, who are interested in the study. For many non-researchers a visual depiction of the procedures can be beneficial to assist in understanding the methods. This is a recommendation and not a requirement.

Author Response

Thank you for reviewing the article again. We have looked at your sugestion and discussed it with between the authors and decided that we think that it is not nescesary to include a figure for showing the protocol. It is also easy to follow in the text. We hope that the reviewer agrees that it is also ok without the figure.

Kind regards